# Poorer Regions Consume More Undeveloped but Less High-Quality Land Than Wealthier Regions—A Case Study

Vlaďka Kirschner [1],*[ID], Daniel Franke [1], Veronika Řezáčová [2] and Tomáš Peltan [1][ID]

1   Department of Landscape and Urban Planning, Faculty of Environmental Sciences, Czech University of Life Sciences Prague, 16500 Prague, Czech Republic
2   Crop Research Institute, 16106 Prague, Czech Republic
*   Correspondence: kirschner@fzp.czu.cz

**Abstract:** Despite the efforts of developed countries to protect undeveloped land, development continues to expand beyond urban boundaries. High-quality land needed for food production is often consumed. This study aims to verify possible causes of undeveloped land and high-quality land consumption within regions (NUTS3) using a new approach to building growth monitoring. It investigates residential (RBs) and commercial buildings (retail and industrial buildings, RIBs). The development between 2006 and 2016 in the Czech Republic, a country in Central Europe, is used as a case study. Population growth and gross domestic product per capita (GDP) within regions are considered two potential causes of land consumption; this hypothesis is verified using a linear regression model. Only GDP showed statistically significant results. It correlated negatively with RBs and RBs + RIBs built on undeveloped land and positively with RBs + RIBs and either RBs or RIBs built on high-quality land. Based on the results, we recommend that land protection policies be differentiated according to regional specifics to be more effective. Regions with lower GDPs should obtain more support in protecting undeveloped land against residential development. The protection of high-quality land should be emphasized by supporting residential and commercial development on brownfield sites in regions with higher GDPs.

**Keywords:** Czech Republic; land consumption; land protection; high-quality land; residential building; commercial building





## 1. Introduction

A need for the protection of land against new development has been recognised in many parts of the world, and the land consumed by new buildings is a significant cause of diversity decline in the world [1]. In the European Union (EU), land protection against urbanisation is perceived as one of the most pressing environmental protection themes [2]. In 2013, the EU established a target of having no land consumed by new development in the EU by 2050 [3], which became known by a short slogan, "no net land take by 2050". However, not much progress on the EU level was recorded as of 2020 [4]; it seems the goal is not being achieved (EEA, 2020). This paper aims to contribute to the debate on how policy can better protect the land in order to achieve this target.

The European Environment Agency [5] monitors the land take, defining it as the 'change of the amount of agriculture, forest and other semi-natural and natural land taken by urban and other artificial land development.' The soil being covered by development results in a loss of biodiversity, which is linked to the physical shrinking of biotopes, and the soil's capacity to regulate the water cycle is heavily compromised [6]. At the same time, the planet's population is increasing, and new migrants may advance the European population in the future [7], increasing the demand for land for living and farming. As land and soil are non-renewable resources, the solution seems to be an efficient use of land, preferably using vacant, so-called brownfield sites, and those with a lower soil quality.

Both the Organization for Economic Co-operation and Development (OECD) and the EU support a compact city policy and the protection of open, undeveloped land [8,9]. Special attention should be given to high-quality agricultural land, which should primarily be used for cultivation [10].

The EU established the target, and the national governments are expected to formulate planning objectives. Some countries, such as Luxembourg, Austria, and Germany, have set national objectives [11]. Some countries, such as Germany, declare that the national objectives are insufficient for their application [12]. There are two reasons for this. Firstly, there are differences between regions and municipalities, so the objectives should be formulated relative to those differences. Germany, for instance, related the objectives to the number of inhabitants within an area (ibid). The second reason regards policy delivery. Regional or local authorities, not the national government, are responsible for delivering planning policies in the EU [6]. As such, some regions should formulate planning policies for dealing with regionally specific demography and land use [13]. In addition, the European Commission [14] highlights the crucial role of regional and local authorities, specifically in addressing soil sealing. Therefore, an evaluation of the factors that might affect growth on undeveloped and high-quality land should be performed on the regional level, using European NUTS3 units (Nomenclature of Units for Territorial Statistics), which are uniform for the EU.

The two main drivers of land consumption in the EU are population growth and economic development [1]. However, the EU target does not distinguish between types of land use [10], i.e., residential and commercial [11]. However, different land uses are affected by different drivers. Moreover, the various drivers may differently affect the new development on both undeveloped and high-quality land; therefore, building on undeveloped versus high-quality land might differ for attractive and economically strong regions. Regional attractiveness can be expressed by population growth and gross domestic product growth (GDP).

We aimed to verify the relationship between residential buildings (RBs) and retail and industrial buildings (RIBs) on undeveloped land and high-quality land and population growth and GDP. Specifically, we estimated the connection between (i) the consumed undeveloped land with RBs, RIBs, and RBs + RIBs, respectively, and the population growths and GDPs in the NUTS3 regions, and (ii) the consumed high-quality land with RBs, RIBs, and RBs + RIBs, respectively, and the population growths and GDPs in the NUTS3 regions (iii). Moreover, we aimed to present a new method for monitoring growth on undeveloped land compared to growth on built-up land.

As a case study, we use the Czech Republic, a developed country located in the centre of Europe, a representant of a post-socialistic country. Here, as well as in Slovakia, Hungary, Slovenia, the Baltic States, and, to a certain extent, Poland [14], land consumption began only in the late 1990s. The development further continued after the expansion of the EU in 2004, when the newly joined countries obtained many subsidies aiming to equalise economic and social conditions in EU countries and their regions [15]. In the Czech Republic, within 20 years of 1989 (the year of the Velvet Revolution), more than 30% of inhabitants had moved outside of the cities [16], which was described in Bičík and Jeleček [17] and Sýkora and Stanilov [14] in detail. Despite a slower pace of building since the economic crisis of 2008 [14], the growth in regions with large cities has remained strong in the Czech Republic [18]. Built-up areas often grow at the expense of agricultural areas [17,19,20]. In the last three decades, a drastic reduction in agricultural activities was recorded, especially in post-socialistic countries [21]. In the Czech Republic, there is no specific national or regional policy objective for limiting growth on undeveloped land in order to fulfil the EU target. However, a strict requirement to protect high-quality agricultural land against new development was launched in 2006 (Construction Act No. 500/2006).

We hypothesised that population growth would positively correlate with the buildings' growth on both undeveloped and high-quality land, and we hypothesised that GDP would negatively correlate with the buildings' growth on both undeveloped and high-quality land.

## 2. Materials and Methods

Data about the growth of RBs and RIBs between 2006 and 2016 on undeveloped and high-quality agricultural land in 14 regions of the Czech Republic were created. Data about population growth and GDP for the same time period and the same regions were calculated in order to test the hypotheses.

Research manuscripts reporting large datasets that are deposited in a publicly available database were used. The accession numbers have not yet been obtained at the time of submission; they will be provided during review.

### 2.1. Study Area

The Czech Republic is administratively divided into 14 regions (see Figure 1), corresponding with European NUTS3 units (Nomenclature of Units for Territorial Statistics). There is, nevertheless, some irregularity in the regional division. The region of the City of Prague (Prague region) is both a region and a municipality, resulting in a lack of undeveloped and notably high-quality land in this region. Therefore, we considered 14 regions for the analysis of undeveloped land and only 13 regions for the analysis of high-quality land, excluding the Prague region. According to the Regional Development Strategy [22], there are three metropolises in the Czech Republic; in addition to Prague, they are Brno (in the South Moravia region) and Ostrava (in the Moravia-Silesia region). We refer to the metropolitan regions for interpreting some phenomena in regional development.

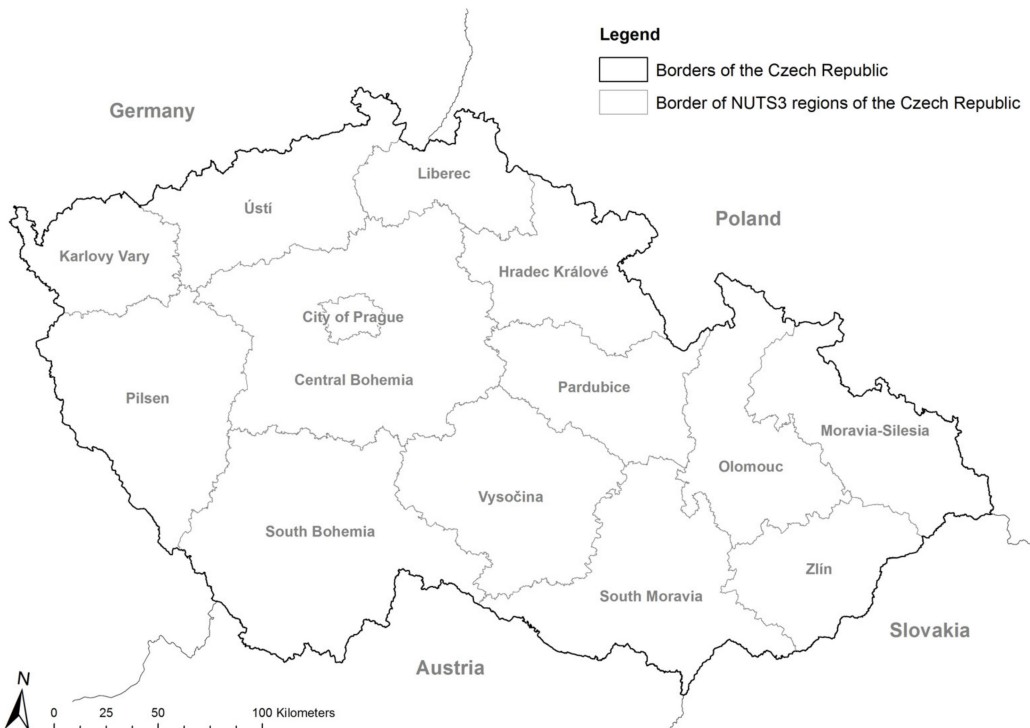

**Figure 1.** The map of the study area. Borders of NUTS3 regions in the Czech Republic (source: Register of Territorial Identification, Addresses and Real Estate, https://www.cuzk.cz/ruian/, accessed on 10 October 2022).

### 2.2. Processing Data on Building Growth

We used the data about buildings that obtained a building descriptive number, collected yearly in the Register of Census Districts and Buildings by the Czech Statistical

Office. This enables more precise analysis compared with the alternative of using data about erected buildings obtained via remote sensing (see the Discussion Section). The data contain information about the area dimensions of the buildings, the way they are used (commercial or residential purposes), and their location. They are in the form of georeferenced points. Such data need to be transformed using a GIS (Geographical Information System). The methodology is summarised in Figure 2.

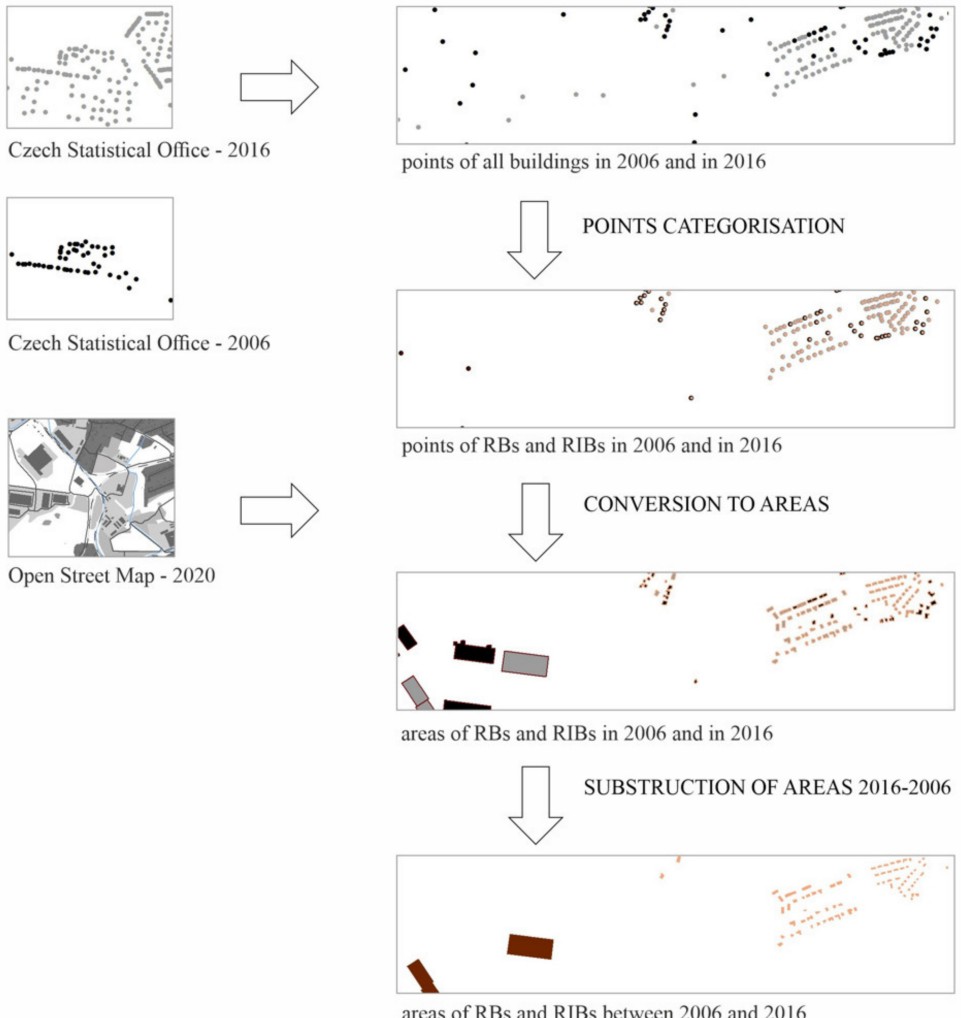

**Figure 2.** Data processing scheme of the building growth between 2006 and 2016.

The used data consist of two sets. The first dataset includes all existing buildings built up until 2006, while the second includes those built up until 2016. Only the Residential Buildings (RBs) and Retail and Industrial Buildings (RIBs) were chosen and joined into one map ("points categorisation"). The RBs consist of the buildings with the following attribute values: *construction with residential use* (code 03), *construction with prevailing residential use* (code 06), and *construction which, by its form, corresponds to a single-family dwelling* (code 07). The RIBs include buildings with the following attribute values: *industrial factories and warehouses* (code 01); *factories with shops and other salerooms, retail spaces, shopping malls, and the like* (code 10); and *factories designed for industry, handicraft, and other production or services of a production nature, and the warehousing of products, staff, and materials* (code 12). Code 01 accounts for 19%, code 10 for 23%, and code 12 for 58% of all the RIBs. If there was no building code in the dataset for the year 2006, the missing code was replaced by the one from the 2016 dataset, assuming that the building's use was not changed over time.

We used the Open Street Map (OSM) to convert points of buildings into areas of buildings by intersecting the building data with the OSM, which includes polygons (areas) of all existing buildings in the year of the map's creation. We used the OSM from 2020 to ensure both 2006 and 2016 buildings were included. The advantage of the OSM is that it is available worldwide and is of sufficient quality. Finally, the building growth was obtained by simply subtracting the area of buildings in 2006 from that of 2016.

The building growth can be divided according to territorial units. In this article, we use the NUTS3 regions. In the Czech Republic, they are freely available from the digital vector geographic database of the RÚIAN (Register of Territorial Identification, Addresses, and Real Estate, https://www.cuzk.cz/ruian/, accessed on 10 October 2022).

### 2.3. Defining Building Growth on Undeveloped Land

We assume the unbuilt area is simply land that does not contain any buildings. There are two possible vector data sources available to identify the land without buildings: Urban Atlas and Corine Land Cover (CLC). Urban Atlas is more precise than CLC [23]; however, it does not cover the entire area of the Czech Republic or the entire EU. We, therefore, use CLC in this study to allow for this methodology to be used in other European countries as well. CLC monitors the land use in 1990, 2000, 2006, 2012, and 2018. We used the dataset of 2006; the analysis can, however, be conducted analogically in the other monitored years.

Undeveloped land (i.e., land without buildings) contains the following CLC layers (including the layer code in brackets): *Arable land* (2.1), *Permanent crops* (2.2), *Pastures* (2.3), *Heterogeneous agricultural areas* (2.4), *Forests* (3.1), *Scrub and/or herbaceous vegetation associations* (3.2), *Open spaces with little or no vegetation* (3.3), *Inland wetlands* (4.1), and *Inland waters* (5.1). It needs to be noted that undeveloped land also contains water and wetlands, where no buildings are constructed.

Finally, the building growth data (Figure 2) were intersected with the undeveloped land layers to define only the buildings built between 2006 and 2016 on undeveloped land.

### 2.4. Defining Building Growth on High-quality Agricultural Land

In order to define high-quality land, we used the Czech database of agricultural land based on the Act for Protection of Agricultural Land (Act No. 334/1992, using a plot) as an alternative to the European Soil Database (using a raster of 1km x 1km). The Act (§1 No. 334/1992) defines agricultural land as agriculturally cultivated land, i.e., "arable land, hops, vineyards, gardens, orchards, permanent grassland". The database is based on a complex and detailed land classification and is regularly updated. Agricultural land is classified into five soil protection classes according to soil quality. In line with the Construction Act (Act No. 500/2006), which protects high-quality soil against new constructions, we used the first two classes with the highest soil protection (Act No. 334/1992) as the high-quality land for the analyses. It must be noted that the agricultural land is monitored regardless of its location (built-up or undeveloped land). Nevertheless, only a minor part of it is located within the built-up land (typically gardens and orchards).

Finally, the building growth data (Figure 2) were intersected with the high-quality land within the agricultural land database to define only the buildings built between 2006 and 2016 on high-quality land.

### 2.5. Analyses

First of all, we needed to calculate comparative values. The GDP variable was calculated as GDP per capita. The buildings' growth on undeveloped land was calculated as a percentage of the buildings' growth anywhere, i.e., RBs growth on undeveloped land is the percentage of RBs growth on undeveloped and built-up land in the region (anywhere), while RIBs growth on undeveloped land is the percentage of RIBs growth anywhere, and RBs + RIBs growth on undeveloped land is the percentage of RBs + RIBs anywhere. Analogically, the buildings' growth on high-quality land was calculated as a percentage of the buildings' growth anywhere.

A simple linear regression analysis was performed with MS Excel data analysis and used to describe the relationship between population growth, GDP growth per capita (independent variables), and undeveloped land consumption and high-quality land consumption (dependent variables). The data were verified by a normality check. As a result, all the variables, except for the RBs and RIBs growth on undeveloped land, were log-transformed before the analyses.

## 3. Results

This section may be divided by subheadings. It should provide a concise and precise description of the experimental results, their interpretation, as well as the experimental conclusions that can be drawn.

### 3.1. Conditions in the Study Area

The results are presented in Table 1. The first two columns show what proportion of land is occupied by undeveloped land and high-quality land in each region. We can see a marked difference between the lands' proportions. While the average percentage of undeveloped land is 90% (SD 12), the average proportion of high-quality land is 24% (SD 8). The difference is even more significant if we keep in mind that this is only the City of Prague (46.4%), with prevalently built-up land, which decreases the average values. In short, the results show that it is high-quality land that is much more limited, and this applies in all regions within the Czech Republic.

**Table 1.** Population growths, GDP per capita growths, and residential (RBs) and commercial (RIBs) buildings' growth on undeveloped land and on high-quality land in the Czech Republic between 2006 and 2016.

| Region | Undeveloped Land in 2006 | High-Quality Land in 2006 | Population Growth | GDP per Capita Growth | RBs Growth | RIBs Growth | RBs + RIBs Growth | RBs Growth | RIBs Growth | RBs + RIBs Growth |
|---|---|---|---|---|---|---|---|---|---|---|
| | | | | | On undeveloped land | | | On high-quality land | | |
| | * | * | ** | | 1 | 2 | 3 | 1 | 2 | 3 |
| | % | % | % | Euro | % | % | % | % | % | % |
| Central Bohemia | 93.3 | 28.7 | 14.6 | 5241.2 | 54.1 | 59.8 | 55.4 | 41.1 | 53.5 | 43.9 |
| City of Prague | 46.4 | 35.3 | 7.3 | 13,970.6 | 30.3 | 41.0 | 32.3 | 33.6 | 50.9 | 36.9 |
| Hradec Králové | 93.3 | 30.8 | 0.6 | 4867.6 | 49.1 | 40.6 | 46.6 | 41.4 | 58.0 | 46.4 |
| Karlovy Vary | 94.9 | 11.4 | −2.1 | 2473.5 | 56.4 | 43.3 | 51.6 | 18.4 | 25.5 | 21.0 |
| Liberec | 93.8 | 18.0 | 2.5 | 3421.0 | 61.2 | 53.8 | 58.1 | 30.6 | 58.7 | 42.6 |
| Moravia-Silesia | 90.4 | 24.4 | −3.0 | 4211.6 | 60.1 | 37.5 | 51.4 | 45.5 | 56.1 | 49.6 |
| Olomouc | 93.4 | 30.8 | −0.7 | 4544.7 | 45.1 | 34.5 | 41.3 | 65.7 | 70.3 | 67.3 |
| Pardubice | 93.3 | 26.5 | 2.0 | 4270.0 | 51.0 | 37.9 | 46.7 | 35.4 | 49.8 | 40.2 |
| Pilsen | 96.4 | 10.8 | 4.6 | 4942.6 | 52.3 | 49.6 | 51.0 | 23.4 | 45.5 | 34.0 |
| South Bohemia | 96.6 | 18.1 | 1.6 | 3332.5 | 57.4 | 46.2 | 54.5 | 31.8 | 38.3 | 33.5 |
| South Moravia | 92.8 | 39.8 | 4.0 | 5543.0 | 41.1 | 38.8 | 40.2 | 61.5 | 79.2 | 68.0 |
| Ústí | 91.4 | 21.2 | 0.0 | 3110.3 | 55.0 | 41.5 | 49.5 | 23.6 | 42.2 | 31.2 |
| Vysočina | 96.1 | 28.3 | −0.3 | 4251.1 | 59.3 | 47.3 | 55.5 | 48.8 | 57.5 | 51.6 |
| Zlín | 92.8 | 17.3 | −0.9 | 5284.3 | 46.4 | 22.1 | 37.2 | 37.0 | 43.7 | 39.5 |

* 100%: regional area. ** 100%: population in 2006. [1] 100%: RBs growth. [2] 100%: RIBs growth. [3] 100%: RBs + RIBs growth.

The population and GDP per capita grew over the ten years under review. While GDP per capita increased in every region, the population increased in only 53% of the regions. Again, the Prague region and its surrounding region (Central Bohemia) were shown to be the most attractive regions for the economy and new inhabitants, followed by the second metropolitan region, the South Moravia region (see Table 1). However, GDP growth does not go hand in hand with population growth by a long sight (e.g., see the Olomouc, Zlín, or Vysočina regions). The least attractive region in terms of population decline (not GDP) was the third metropolitan region, the Moravia-Silesia region.

The buildings' growth on undeveloped land and high-quality land are displayed in Figure 3, where the values are arranged alphabetically by region. The graphs clearly show

more variability in the growth on high-quality land (43 ± 13%) than on undeveloped land (47 ± 7%). Another marked difference regards the use of the buildings. While RBs are more often built on undeveloped land (RBs: 51 ± 8%; RIBs: 42 ± 9%), RIBs are more often built on high-quality land (RIBs: 53 ± 13%; RBs: 41 ± 8%). Such difference is particularly notable in the Moravia-Silesia or Zlín regions (for growth on undeveloped land) and in the Liberec or Pilsen regions (for growth on high-quality land). The South Moravia region has, markedly, the highest proportion of RIBs on high-quality land.

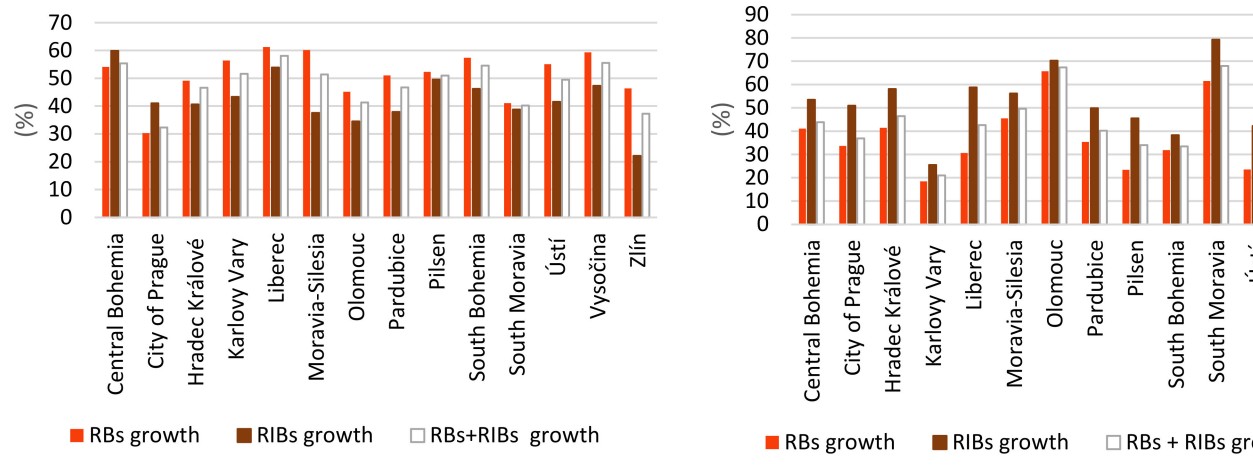

(**a**) Growth on undeveloped land.　　　　　　　　　　　(**b**) Growth on high-quality land.

**Figure 3.** Percentages of buildings (RBs, RIBs, and RBs + RIBs) built on undeveloped land (**a**) and high-quality land (**b**) in the Czech Republic between 2006 and 2016. 100% = buildings built anywhere.

*3.2. Relationship between Population Growth/GDP Growth and Buildings' Growth on Undeveloped/High-quality Land*

The regression analysis was performed to explain the relationship between population growth and the percentages of the buildings' growth on undeveloped land and high-quality land, respectively. The analysis did not significantly ($p = 0.05$) explain any of the relations. However, on $p \leq 0.1$ (specifically, $p = 0.06$), the RIBs growth on undeveloped land was positively correlated with the population growth. The model, however, explained 27% of the data variability.

The other regression analysis was performed to explain the relationship between the GDP growth per capita and buildings' growth on undeveloped land and high-quality land, respectively. We compared only 13 regions in the high-quality land model, excluding Prague's region, since Prague contains an abundance of quality soils, and these soils are, unlike in other regions, primarily located in the built-up land (Table 1). This model (displayed in Figure 4) explained most of the relations with statistical significance. The model explains approximately 50% of the data variability in the relationship between GDP and all (RBs + RIBs) buildings built on undeveloped land or high-quality land, and between 45 and 74% of the data variability in the relationships between GDP and RBs built on undeveloped and high-quality land, and RIBs built on high-quality land (Figure 4). However, the relationship between GDP and the buildings' growth on undeveloped land is negative, while the relation between GDP and high-quality land is positive. Only the relationship of GDP with RIBs built on undeveloped land was insignificant, at $p \leq 0.05$. We can interpret this as follows: if GDP grows in a region, RBs consume relatively less undeveloped land in that region (explaining 74% of the variability—4b). Conversely, if GDP grows in a region, both RBs and RIBs consume relatively more high-quality land (explaining 45% and 49%, respectively, of the variability—4e and 4f). We cannot explain with significance the relation between GDP growth and RIBs growth on undeveloped land ($p = 0.06$) (Figure 4c).

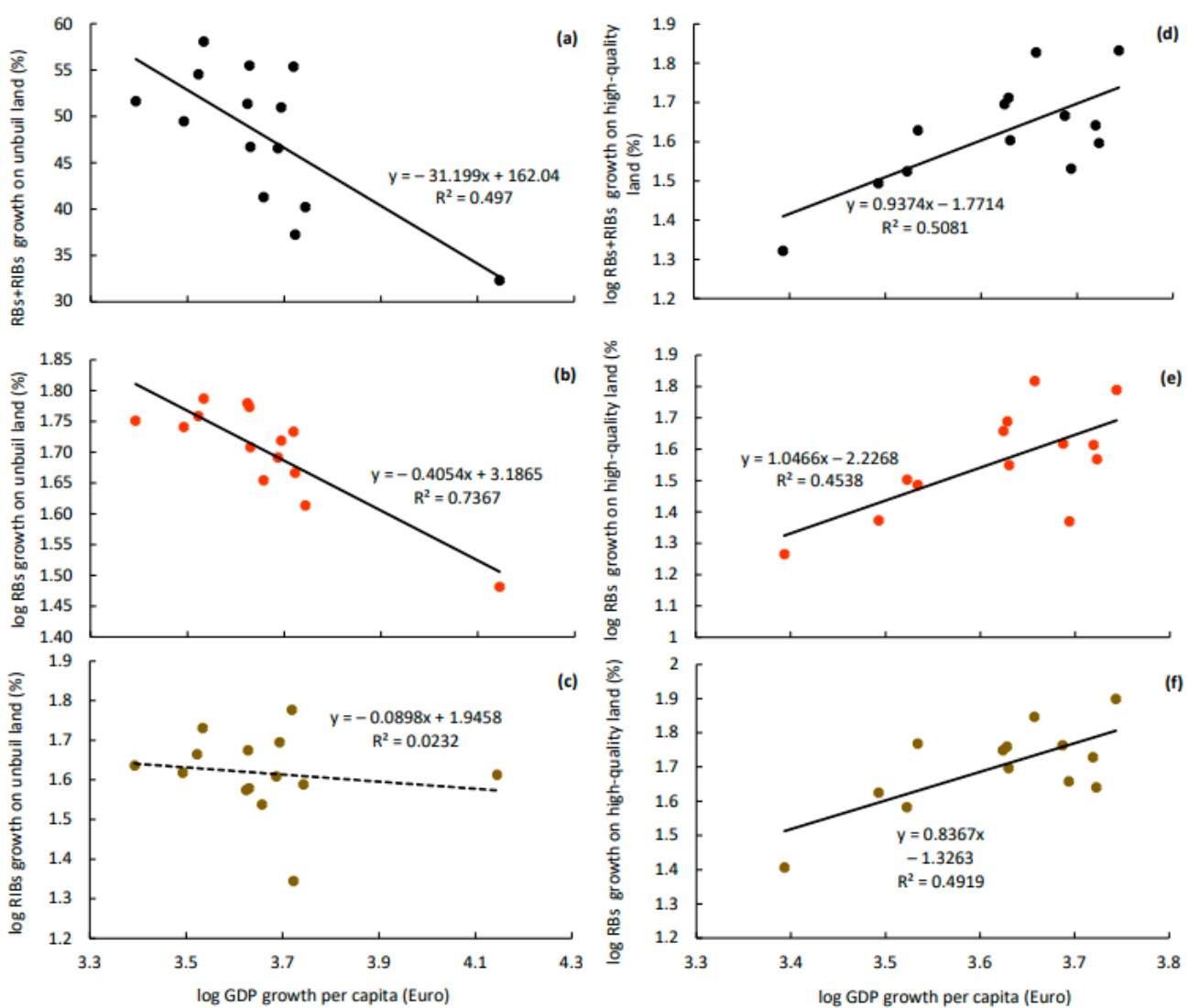

**Figure 4.** Plots of linear regression analyses showing the relationship between GDP per capita and building growth on undeveloped land (**a–c**) and high-quality land (**d–f**). The dashed line in the trend line represents a statistically insignificant (on $p \leq 0.05$) result (**c**).

## 4. Discussion

### 4.1. The Monitoring of Building Growth on Undeveloped Land

The presented method of processing data on building growth (Chapter 2.2) can be used as an alternative method for the same data derived from remote sensing monitoring [5,13,24–26]. Remote sensing is a widely used method; however, it does not detect small-scale development [27], including the precise resolution of buildings' uses [23]. Nevertheless, most of the current development on undeveloped land in the EU is small-scale, scattered development [26,28], and growth on undeveloped land most frequently comes in the forms of residential and commercial development [1]. The method is undoubtedly limited to countries with a database of existing or newly built buildings. However, most if not all developed countries own such data. The method of defining buildings constructed on undeveloped and built-up land (Chapter 2.3) presented here can thus be applied in all developed countries, if not globally. We compare the ratio of the growth on undeveloped land to the whole, not the amount of growth on undeveloped land. The ratio comparison might support a policy target reflecting the compact city policy.

*4.2. Trends in Land Consumption*

According to the EU database [5], the Czech Republic is among the average countries in the EU in terms of land consumption. Just as in many EU countries [5], about half of the development here occurred on undeveloped land between 2006 and 2016. As such, the Czech Republic and many other EU countries [11,29] are not on their way to reaching the 'no land take' EU target. There is undoubtedly a need to target the planning tools to protect undeveloped land more efficiently. Based on our results, the policy protecting undeveloped land could primarily target residential development. In the long term, there has been a noticeable decrease in agricultural land in the Czech Republic [24,30], as in other post-socialistic EU countries [15,25], caused by building growth and afforestation (ibid). Our results further indicated a decline in high-quality agricultural land from 2006 to 2016; high-quality land is also much more limited than undeveloped land in the Czech Republic. Therefore, we suggest that the policy protection of high-quality land should be better emphasised. The policy should focus on both types of development. On the one hand, RIBs consume a relatively higher share of high-quality land, but those higher values might be caused by the scale of development, due to the need for large tracts of mostly flat land, which often tend to be high-quality agricultural land. On the other hand, RBs should not be omitted because of a lower share, as this type of development is less sensitive to terrain configuration and location. The policy should be highly regionally differentiated.

Our study did not confirm the relationship between population growth and growth on undeveloped or high-quality land. Although population change does not cause the buildings' growth, RB growth in regions with population decline is not in any case an efficient use of land. We did confirm a paradoxical EU trend that the buildings' growth occurs despite population decline [15,31]. This trend was especially noticeable in post-socialistic countries [32] with a concentration of high-density housing estates in urban peripheries [14]. People's desire for individual housing was the initial impetus for housing expansion beyond the city boundary after the fall of the communist regime. Such a trend continues especially in post-industrial regions (primarily the Moravia-Silesia region) with an exceptionally high concentration of working-class housing estates. People are either moving to individual housing in the city outskirts, while the population in the core city (Ostrava) declines [31], or leaving the whole region towards working opportunities in economically wealthier regions [33]. The same trends were observed in the adjacent Silesia region in Poland [32], eastern Germany [34], and Hungary [35].

Nevertheless, we identified a slight correlation between population growth and RIBs growth on undeveloped land. A possible explanation is that an influx of new inhabitants increases congestion in built-up areas, pushing "unattractive" commercial buildings out of the urban boundaries. Such an explanation is in contradiction with the bid rent theory [36], which says that inhabitants are pushed out by industry. Nevertheless, this theory is based on the American experience in the 1960s. Another explanation might be that the slight correlation is caused by the exceptionality of the Prague region. At the beginning of 2000, commercial centres were built beyond the administrative borders of Prague because EU subsidies for these centres could not be obtained for Prague itself due to its high GDP per capita; therefore, the slightly poorer Central Bohemian Region surrounding Prague obtained the subsidies [18]. Consideration must be given as to whether Prague should be regarded together with the Central Bohemia region for similar cases. European Integrated Territorial Investments might be the tool to overpass the issue.

There was a contradictory effect of GDP growth on the buildings' growth on high-quality land and on undeveloped land. The difference in the results supports the approach of considering growth on undeveloped land and on high-quality land separately at the EU level [37], and, consequently, on national and regional levels. The RIBs growth on undeveloped land does not reflect the level of GDP in the region. This is probably caused by the fact that profit-generating companies are not always based in the same region as the one where they engage in business, i.e., where they build RIBs. RIBs are more often built on high-quality land than RBs, growing more on high-quality land in prosperous

regions. The reasons may lie in a combination of operational and historical reasons. The majority of RIBs (77%—codes 1 and 12) are buildings requiring truck traffic for service. As such, they are concentrated along highways and important transport intersections [38] (primarily undeveloped land), especially near large cities [18]. Such locations probably have high-quality land because prosperous regions with large cities can often be traced back to successful medieval settlement structures with a strong dependence on high-quality agricultural land (the path-dependence theory [39]).

With increasing GDP, RBs grow relatively more on high-quality land, and, at the same time, they grow relatively less on undeveloped land. As high-quality land is situated both on undeveloped land and within built-up areas, we can interpret it as the built-up areas in wealthy regions tend towards greater compactness. This is particularly valid for metropolitan regions (such as with Brno, but not with the structurally affected Ostrava). Such a trend probably reflects a change in the living preferences of people who moved to wealthier regions for work. Low-density suburban housing with greater demands on daily commuting do not meet their needs [40]. Economic reasons can also affect densification trends; land prices are under higher pressure in wealthier regions with population growth. Planning tools could take advantage and become stricter in limiting growth on undeveloped land. Economic reasons probably become more significant during economic crises, such as after 2008. In fact, suburbanisation during the economic crisis in the Czech Republic decreased [18]. Instead of building houses on undeveloped land, the houses in some post-socialistic countries began to be placed in the gaps within the suburban developments of big cities, thus densifying the existing housing estates [41]. The phenomena can be called "inner" suburbanisation [41] (not to be confused with negative trend suburbanisation "within city limits" [42]). We also identified densification trends in the Czech Republic (approximately 53% ± 7% of RBs occurred on built-up land), and these trends increase in wealthier regions.

In general, the densification trends are in line with the EU policy of sustainable development [8]. However, a negative side effect is a loss of urban greenery with all its positive effects, including broader environmental [43] and social effects, such as recreation and urban agriculture possibilities for inhabitants [28]. Densification should instead take place at the expense of brownfield sites (previously used sites, usually with a lower soil quality) present in every region (a database of brownfields in the Czech Republic can be found at czechinvest.org). The policy must, therefore, primarily support poorer regions with undeveloped land protection. Soil cleaning and brownfield regeneration for RBs should be emphasised within the built-up areas of wealthier regions.

### 4.3. Study Limitations and Further Research

This study finds associations between the variables, not the mechanism behind them. We present several possible explanations of the results, which should be tested in future research. In particular, the negative relation between per capita GDP growth and the share of new residential development on undeveloped land should be explored in more detail. Another issue is the mechanisms behind the consumption of high-quality agricultural land in regions with higher and lower GDP growths. More in-depth research should focus on variables such as the character of recent built-up land (e.g., the density and densification potential) and planning-related variables (e.g., developable land and its characteristics). These variables might differentiate between economic and planning factors in the explanation of the relations described.

The Czech Republic is a case study, and the results will probably not be universally valid. Nevertheless, we assume the results will be valid for all post-communist countries, as the growth tendencies are similar in this region [14,15].

## 5. Conclusions

This study presents a complex and detailed alternative methodology for monitoring land consumption in the EU with the aim of establishing effective planning tools. This study

verified population growth and GDP as two highly potential causes of RB and RIB growth on undeveloped and high-quality land. However, GDP was shown to be highly statistically significant, increasing building on undeveloped land and decreasing building on high-quality land with its growth. Based on the results, we recommend that land preservation policies be differentiated according to the NUTS3 units to propose a spatially oriented policy that reflects regional differences. The regions with lower GDPs should obtain more support to protect undeveloped land against residential development. The protection of high-quality land should be emphasised by supporting residential and commercial development on brownfield sites in regions with higher GDPs. Last but not least, this study presents the amount and distribution of RBs and RIBs across the regions (NUTS3) of the Czech Republic between 2006 and 2016, thus contributing to the worldwide debate on land preservation for future generations.

**Author Contributions:** Conceptualization, V.K. and V.Ř.; methodology, D.F.; software, D.F.; validation, T.P. and D.F.; formal analysis, V.Ř. and V.K.; investigation, V.K.; resources, V.K. and T.P.; data curation, D.F.; writing—original draft preparation, V.K.; writing—review and editing, V.Ř. and V.K.; visualization, V.K.; supervision, V.Ř.; project administration, V.K.; funding acquisition, V.Ř. All authors have read and agreed to the published version of the manuscript.

**Funding:** This research was funded by the Ministry of Agriculture of the Czech Republic, grant number MZe RO0418.

**Data Availability Statement:** Not applicable (All data are published within the article. No other data are published elsewhere).

**Conflicts of Interest:** The authors declare no conflict of interest.

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
