# Peer review of "Poorer Regions Consume More Undeveloped but Less High-Quality Land Than Wealthier Regions—A Case Study"

_land, doi:10.3390/land12010113_

Round 1

Reviewer 1 Report

If you have available data add the next five years to the analysis (2017, 2018, 2019, 2020, 2021).

Author Response

Dear Reviewer,

Thank you for your very kind comments.

Unfortunately, we do not have the data available from 2017 – 2021. We got ten years of data as part of a project from the Czech Statistical Office. In this comprehensive and spatial form, this data is available for the purpose of a specific project. The data are also publicly available from the digital vector geographic database of the RÚIAN (Register of Territorial Identification, Addresses and Real Estate). However, they must be withdrawn town by town (https://www.cuzk.cz/Katastr-nemovitosti/Poskytovani-udaju-z-KN/Vymenny-format-KN/Vymenny-format-NVF.aspx#k1). They are 6254 towns in the Czech Republic. Unfortunately, the withdrawal cannot be automatised using a batch tool.

We believe this explanation would be to your satisfaction.

With kind regards,

Authors

Reviewer 2 Report

The article presents a case study related to the development of agricultural land and the need to protect agricultural land and the policy implemented by the EU 'not net land take by 2050' and the reasons for little progress in this direction. New ways of monitoring the new state of agricultural land development are also presented. The research covered 14 regions excluding the Prague region due to the specificity of these areas. 

The paper is written correctly in terms of the structure of the scientific article. The abstract is briefly written and reflects the content of the research described in the article, but could be developed further.  The literature review is also positive, referring directly to the subject of the research and presenting mainly the current scientific achievements in this field, although it should be expanded to include international literature. In the reviewer's opinion, the methodology of the conducted research is insufficiently clearly described. Minor linguistic and spelling errors. 

Summing up: The work meets the requirements of the publisher only to a sufficient degree. The aim of the work is clearly defined, but its description is not clear enough in the reviewer's opinion. In the reviewer's opinion, the work meets the requirements of the journal to a minimum degree and can be published. 

Author Response

Dear Reviewer,

Thank you for your kind comments.

In response to your comments, we have added more international literature and stylistically modified the methodology to make it more straightforward for the reader. The methodological changes can be found in chapters 2.1 – 2.5.

Internation literature relevant to countries from Central and Eastern Europe is discussed in the Discussion section (lines 327-337, and 389-395). In particular, we added the following international sources:

  1. Haase, A.; Athanasopoulou, A.; Rink, D. Urban Shrinkage as an Emerging Concern for European Policymaking. Eur Urban Reg Stud 2016, 23, 103–107, doi:10.1177/0969776413481371.
  2. Nuissl, H.; Rink, D. The "production” of Urban Sprawl in Eastern Germany as a Phenomenon of Post-Socialist Transformation. Cities 2005, 22, 123–134, doi:10.1016/j.cities.2005.01.002.
  3. Kovács, Z.; Farkas, J.Z.; Szigeti, C.; Harangozó, G. Assessing the Sustainability of Urbanization at the Sub-National Level: The Ecological Footprint and Biocapacity Accounts of the Budapest Metropolitan Region, Hungary. Sustain Cities Soc 2022, 84, doi:10.1016/j.scs.2022.104022.
  4. Sýkora, L.; Ouředníček, M. Sprawling Post-Communist Metropolis: Commercial and Residential Suburbanisation in Prague and Brno, the Czech Republic. In Employment deconcentration in European Metropolitan Areas: market forces versus planning regulations; Razin, E., Dijst, M., Vázquez, C., Eds.; Springer: Dordrecht, 2007; pp. 209–233.
  5. Spórna, T.; Krzysztofik, R. ‘Inner’ Suburbanisation – Background of the Phenomenon in a Polycentric, Post-Socialist and Post-Industrial Region. Example from the Katowice Conurbation, Poland. Cities 2020, 104, 102789, doi:10.1016/j.cities.2020.102789.
  6. Vasárus, G.L.; Lennert, J. Suburbanization within City Limits in Hungary—A Challenge for Environmental and Social Sustainability. Sustainability (Switzerland) 2022, 14, doi:10.3390/su14148855.

We believe the changes would be to your satisfaction.

With kind regards,

Authors

Reviewer 3 Report

Authors of the article presented an interesting research approach, aiming to verify possible causes of undeveloped land and high-quality land consumption within regions (NUTS3) in Czech Republic. Using linear regression model they analyzed in sufficient, correct methodologically way the patterns of changes in land use by residential and commercial development. That is the biggest methodological advantage of the material given for the review. In my opinion the originality and quality of presented data and methods is high.

As Authors are arguing: ‘with increasing GDP, RBs grow relatively more on high-quality land, and, at the same time, they grow relatively less on undeveloped land. This effect can be caused by  both larger development pressure on open land and by the high quality of agricultural land, such as gardens and orchards, on built-up land. Path-dependence [33] can also play a role: successful regions can often be traced back to successful medieval settlement structures with a strong dependence on high-quality agricultural land.’ (lines 330-335) I would also say that the RBs are more likely built/developed in the big cities surroundings than in the distant, peripheral locations and, consequently, less likely in the regions that have relatively low GDP per capita. There are higher investor tendencies to develop new housing zones closely to the most promising agglomerations (where the high-quality land is still available and on the other hand the housing market more widely reflects the suburbanization processes, clients are willing to look for the new attractive locations, usually qiute highly-densed, with good transportation infrastructure and small distances from the core city). What I want to suggest is that in the article there is a lack of three possible and interesting perspectives i.e.:

1)      To enter into the discussion the context of clients preferences on the real estate markets (both housing and commercial) and its influence on the phenomenon investigated in the paper.

2)      Spatial interdependencies which could or should bring Authors to the thinking about scale and role of the main cities in the analyzed regions and about their influence on the regional economic development in general.

3)      The deeper analysis of types of regions considered in the research, the characteristics and dominant directions/specifics of development could bring Authors to the classification of certain types of land consumption (but such the approach would probably need considerable increasing of the numer of regions in research sample). This could lead to taking into account also other CEE countries.

These additional perspectives would probably allow to formulate more complete reference to the results of research in the final parts of the paper (meaning: discussion and conclusions). Potential similarities and / or differences between regions in Czech Repuplic (and other CEE countries) might be an attractive comparison platform enabling a deeper inference about the determinants of investigated phenomenon. Generally Authors should present a broader explanation and find the way to improve arguing the effects on land consumption shown in the article. In my opinion after such supplementation the paper is worth of publication.

Author Response

Dear Reviewer,

Thank you for your kind and helpful comments. We have integrated the required changes to the manuscript.

  1. To enter into the discussion the context of clients preferences on the real estate markets (both housing and commercial) and its influence on the phenomenon investigated in the paper.

We added the context of clients´ preferences on the real estate markets including its influence on the investigated phenomenon into the discussion.

Clients' preferences on the housing market – in the Discussion section, lines 328 – 336, and 382-386.

Clients' preferences on commerce – in the Discussion section 359 – 365, add a note to the methodology – line 145 – about the proportion of industry and retail.

  1. Spatial interdependencies which could or should bring Authors to the thinking about scale and role of the main cities in the analyzed regions and about their influence on the regional economic development in general.

The differences between the regions and cities are important issues we discussed before the submission. We focused our article aiming only on the relationship between population growth / DPD and the buildings' growth, not the spatial interdependencies. The reason was not to tire the international reader with our regional differences. However, we agree that the article lost the depth of detail.

We, therefore, added notes about some regions and cities into the Methodology section (lines 118-121), Results (lines 235-238 and 246-249), and Discussion (lines 332, 363, 378). In doing so, we have tried to generalize the information as much as possible (using the terms metropolitan cities and regions, wealthy regions, or post-industrial regions) to make the text interesting to an international reader.

  1. The deeper analysis of types of regions considered in the research, the characteristics and dominant directions/specifics of development could bring Authors to the classification of certain types of land consumption (but such the approach would probably need considerable increasing of the numer of regions in research sample). This could lead to taking into account also other CEE countries.

We outlined the issue of types of regions (lines 389-395). However, we agree that it would need considerable increasing of the number of regions over the 14 Czech Regions to create a land consumption typology, and we do not work in the appropriate scale [such as 1, 2]. Rather than creating a typology, we looked for the relations between the variables. Nevertheless, we added some relevant international comparisons to the Discussion section (especially lines 328-338, 349, 389-395).  

In addition, we stylistically modified the methodology to make it more straightforward for the reader.

We believe the changes would be to your satisfaction. Otherwise, we are prepared to take another round of revisions if it will be necessary.

With kind regards,

Authors

Sources:

[1] Spórna, T.; Krzysztofik, R. ‘Inner’ Suburbanisation – Background of the Phenomenon in a Polycentric, Post-Socialist and Post-Industrial Region. Example from the Katowice Conurbation, Poland. Cities 2020, 104, 102789, doi:10.1016/j.cities.2020.102789.

[2] Kubeš, J.; Ouředníček, M. Functional Types of Suburban Settlements around Two Differently Sized Czech Cities. Cities 2022, 127, doi:10.1016/j.cities.2022.103742.
